# Biofeedback Training in Inpatient Mental Health Facilities: A Scoping Review

**DOI:** 10.3390/jcm14103491

**Published:** 2025-05-16

**Authors:** Kira Schmidt, Maike Schlicht, Lina Deutschendorf, Lena Smets, Alexander Bäuerle, Martin Teufel

**Affiliations:** 1Clinic for Psychosomatic Medicine and Psychotherapy, LVR-University Hospital Essen, University of Duisburg-Essen, 45147 Essen, Germany; kira.schmidt@lvr.de (K.S.); maike.schlicht@lvr.de (M.S.); lina.deutschendorf@lvr.de (L.D.); lena.sophie.91@t-online.de (L.S.); martin.teufel@uni-due.de (M.T.); 2Center for Translational Neuro- and Behavioral Sciences (C-TNBS), University of Duisburg-Essen, 45147 Essen, Germany

**Keywords:** BFB, neurofeedback, EEG biofeedback, electromyography, heart rate variability

## Abstract

**Background**: Biofeedback (BFB) has long been a successful treatment for various mental health disorders. The purpose of this scoping review is to investigate the implementation of BFB in inpatient treatment concepts for the therapy of mental health disorders. **Methods**: Through a systematic search via Medline, PubMed, and the Web of Science, as well as a manual search in Google Scholar and reference lists, relevant articles published up to 30 December 2024 were identified. Studies were included if they focused on BFB interventions to treat mental health disorders in inpatient settings and were published in English or German. Studies were assessed by two independent raters, and key information was summarized in a shared document. **Results**: This scoping review analyzed 20 articles published between 1979 and 2022, examining BFB in inpatient settings for various mental health disorders, i.e., obsessive–compulsive disorder, depression, anxiety, substance use disorders, schizophrenia, and eating disorders. Positive outcomes were observed in symptoms, stress reduction, and improvements in cardiac autonomic and motor functions. The duration and frequency of the sessions varied widely, and different methodologies were used across studies, including controlled sessions and self-administered exercises. **Conclusions**: Most BFB inpatient studies showed positive effects on clinical symptoms. There was a broad heterogeneity of the studies. Comparisons are limited, making it challenging to give general recommendations for BFB implementation. The issue remains whether a methodologically consistent approach is necessary for clinical success.

## 1. Introduction

Biofeedback (BFB) has been used for many years in the treatment of various mental health disorders. In BFB therapy, physiological signals such as heart rate, blood pressure, muscle tension, or brain frequency are first measured, amplified, and then presented to the patient as a perceivable stimulus. This measurement is conducted using specialized equipment, with the data processed through dedicated software. The feedback is typically provided through visual or auditory cues, enabling the patient to adjust their physiological parameters consciously. The overarching goal of BFB is to facilitate the modulation of psychological functions by altering physiological processes, ultimately leading to symptom reduction or improved performance [1]. BFB is categorized as a behavioral therapy approach within psychotherapy, following a structured, goal-oriented treatment plan aimed at promoting positive changes in specific behaviors [2]. Several types of BFB exist, such as heart rate variability (HRV) BFB, electromyographic (EMG) BFB, or electroencephalographic (EEG) BFB. HRV BFB is a technique that enables individuals to regulate the variability and dominant rhythms of their heart activity [3]. This type of BFB training aims to enhance the total HRV within a specific frequency range. Research has explored its applications in the treatment of various medical and psychological conditions, including anxiety disorders, asthma, cardiovascular diseases, irritable bowel syndrome, chronic fatigue, chronic pain, and fibromyalgia [3]. EMG BFB provides individuals with real-time feedback on the electrical impulses generated during muscle contractions, allowing them to control and modify their muscle activity [4]. Conscious pain management may be conditioned using this method [5]. EEG BFB modifies brain activity by combining cognitive regulation mechanisms with brain stimulation concepts [2]. Real-time processing of EEG data, the extraction of key parameters, and visual or aural representation of the results constitute this non-invasive brain training technique. Physiological response control and classical and operant conditioning methods may be used to teach behavioral changes in the body’s natural processes, which can change the amplitude of particular frequency bands [2,6]. Brain frequency bands used for training include infra-low frequency, alpha, beta, alpha/theta, high beta, slow cortical potential (SCP), and sensorimotor rhythm (SMR).

However, the success of BFB interventions depends on numerous factors. Beyond the learned modification of physical functions, the potential shift in perceived self-efficacy may also play a significant role in treatment outcomes [7]. It is suggested that the greater the patient’s sense of control over previously involuntary bodily functions, the more likely the therapeutic success will be. Other important factors that contribute to the success of BFB treatment include improved interoception (the awareness of one’s internal bodily processes), positive treatment expectations, and a therapeutic environment free from anxiety. An anxiety-free setting prevents stress reactions that could interfere with physiological measurements, such as muscle tension, changes in respiration, or an elevated heart rate. Classical and operant conditioning mechanisms, along with physiological response control and cognitive mediation, are employed to teach patients how to change their body’s internal processes [2]. By rewarding desirable behavior with intrinsic or extrinsic outcomes, BFB reinforces that behavior, making it more likely to be repeated. The selection of which physiological parameters and training methods to use for alleviating a patient’s symptoms is generally made collaboratively, based on the patient’s specific needs.

BFB is one of several treatment options available for a variety of mental health disorders. Positive effects have been observed in the treatment of depression, generalized anxiety disorder, panic disorders, tension headaches, migraines, insomnia, post-traumatic stress disorder (PTSD), binge eating disorder (BED), and specific phobias [2]. BFB has proven effective in enhancing self-efficacy in patients with depression, leading to significant reductions in both psychological and physical symptoms [8]. For individuals with anxiety and panic disorders, BFB has been shown to decrease heart reactivity to stress and significantly reduce both trait and state anxiety [9]. In the treatment of chronic pain disorders, BFB has resulted in reduced pain and disability [10]. It has also demonstrated effectiveness in alleviating PTSD symptoms, helping to lower nervous system arousal [11,12]. Additionally, BFB has been found to reduce binge eating episodes and emotional eating in BED patients while increasing their ability to manage food-related stress [13]. Particularly notable is the success of BFB in treating attention-deficit/hyperactivity disorder, where it has significantly reduced core symptoms such as inattention, impulsivity, and hyperactivity [13,14].

Despite these positive outcomes, most studies to date have focused on outpatient treatment for mental health disorders, resulting in a significant body of research on outpatient BFB interventions. As a result, there is a notable lack of research on the application of BFB in inpatient mental health settings. A mixed-methods evaluation of BFB treatment in a psychosomatic–psychotherapeutic inpatient unit revealed several implementation challenges, ranging from technical and organizational issues to scheduling conflicts and staff shortages, which often led to canceled sessions [11]. The increased workload and added responsibilities were found to be the most significant barriers. Nevertheless, both patients and clinicians expressed high levels of acceptance and satisfaction with the inpatient BFB program. These challenges underscore the importance of careful planning and organization when implementing BFB interventions in an inpatient setting.

Currently, there is a paucity of studies examining the use of BFB in inpatient mental health facilities. The evaluation mentioned earlier emphasized various obstacles to implementing this treatment, yet it also revealed the substantial potential benefits BFB offers in an inpatient context, particularly in terms of improving patient outcomes.

### Objectives

This review aims to explore the use of biofeedback (BFB) in inpatient settings for treating mental health disorders, marking the first review to specifically address this topic. It seeks to synthesize existing evidence on BFB treatments in such environments. The research questions focus on identifying the types of BFB treatments implemented in inpatient care and evaluating their efficacy. The goal is to provide a comprehensive summary of various studies on BFB interventions, offering a broad overview without narrowing focus on specific treatments, outcomes, or patient groups. If possible, the review will provide recommendations for integrating BFB into inpatient treatment protocols to improve patient care.

## 2. Methods

This scoping review was conducted in accordance with the Preferred Reporting Items for Systematic Reviews and Meta-Analyses Protocols Extension for Scoping Reviews (PRISMA-ScR) guidelines [15]. The aim of this review was to explore the key concepts and limitations in the existing literature on the topic of interest. A scoping review was selected due to the diversity of study designs, methodologies, and research areas, as well as the flexibility it offers in adapting search terms after an initial systematic search. Relevant publications were classified based on the trained BFB parameters (e.g., heart rate variability, muscle tension, brain frequency, etc.). The protocol for this review was preregistered in the Open Science Framework (OSF Digital, Québec, QC, Canada); DOI: 10.17605/OSF.IO/EP2QY).

### 2.1. Literature Search

To identify relevant studies, a systematic search was conducted through Medline, PubMed, and the Web of Science from 1 March 2024 up until 30 December 2024. The initial search focused on PubMed to identify pertinent articles and to develop an appropriate search strategy based on the text words found in abstracts and the index terms used in the articles. The final search string used in PubMed was “neurofeedback”, “biofeedback”, “EEG biofeedback” AND “inpatient” AND “mental health” OR “psych*”. The search terms and search string were then adapted for the other databases accordingly. Additionally, a manual search was performed in Google Scholar to identify any potentially missed studies. Reference lists from relevant articles and previously published reviews were also manually checked for additional studies. Only studies involving adult patients were included.

### 2.2. Study Selection

Inclusion and exclusion criteria were defined a priori. A study was eligible for inclusion in this scoping review if it met the following criteria:

Peer-reviewed article: The article was published in a peer-reviewed journal.

BFB intervention: The study conducted any kind of BFB intervention, including EEG BFB, EMG BFB, and HRV BFB.

Inpatient setting and mental health: Articles were included if they focused on mental health within inpatient settings, encompassing both planned and discharged inpatients, as well as studies from multiple settings that also considered inpatients.

Study type: Due to the exploratory character of this scoping review, different study types were included: randomized controlled trials; non-randomized controlled trials; controlled before and after studies; and case control, cohort, or descriptive studies.

Language: Due to the authors’ language limitations, only articles in English and German were considered.

The first and second authors (KS and LS) of this study independently determined whether an article was eligible for this review according to the abovementioned inclusion criteria. During the initial screening, both raters assessed the relevance of each article by reviewing its abstract and full text. In cases of disagreement between the two raters, a third rater (MS) was consulted, and a consensus was reached through discussion. Reports that overlapped or had duplicate publications were not included in the analysis. A summary of the study selection is presented in Figure 1.

### 2.3. Data Charting

The included studies were extracted in tables by two persons independently (MS, LD). First, the relevant studies were categorized according to the applied type of BFB (either EEG BFB, HRV BFB, or EMG BFB). The data charting process involved recording the following details: the author names, article title, publication year, journal name, country of data collection, type of intervention (trained parameters and number and duration of sessions), key findings, any positive or negative reception of BFB noted in the article, limitations discussed in the study, and suggested future directions. Data charting was performed using an Excel (Version 16.0.5495.1000, Microsoft, Redmond, WA, USA) spreadsheet shared among all researchers, which is included in the Appendix A. The studies were organized according to the applied type of BFB.

## 3. Results

### 3.1. Overview

The search resulted in a total of 168 hits (Figure 1). After excluding duplicates, 143 publications remained. With this remaining number of studies, the publications were checked for relevance and exclusion criteria. As a result, 102 studies were excluded. Ultimately, 20 articles published between 1979 and 2022 were identified for the review. Of the included studies, most were conducted in the United States of America (40%). The number of studies by country and year are summarized in Figure 2. The included studies employed various study designs, including non-randomized (10%), randomized controlled (30%), prospective (5%), pilot (20%), sham-controlled (5%), double-blinded (5%), experimental (5%), crossover (5%), and retrospective approaches (5%). The sample sizes varied across all included studies (range *n* = 10–121; average sample size *n* = 23.74), and a variety of diagnoses were considered.

The effectiveness of BFB in the inpatient setting was demonstrated in 12 out of 20 studies for the following disorders: depression (Beckham et al. [16] and Tatschl et al. [17]), anxiety disorder and depression (Bhat [18]; Cheon et al. [19]), alcohol use disorder (Denney et al. [20]; Penzlin et al. [21] and Teeravisutkul et al. [22]) substance use disorders (Scott et al. [23]; Eddie et al. [24]), obsessive–compulsive disorder (Kopřivová et al. [25]), and schizophrenia (Pharr & Coursey [26]; Cheng et al. [27]). Of these studies, only two reported effect sizes of η_p_^2^ = 0.53 for the effect of HRV BFB on depression (Tatschl et al. [17]) and Cohen’s d = 0.35 for the effect of HRV BFB on substance abuse disorders (Eddie et al. [24]). Since most of the included studies did not report any effect sizes, no meta-analytic investigations were conducted in this study (Figure 3).

### 3.2. Classification

The 20 articles were classified into the following categories: nine articles (45%) used different variations of EEG BFB (infra-low frequency, alpha, beta, alpha/theta, high beta, SCP, or SMR) as treatment during the intervention, seven of the included studies (35%) applied heart rate variability (HRV) BFB, and four articles (20%) implemented electromyography (EMG) BFB during the intervention. Out of the nine EEG BFB studies, four studies (44.4%) chose alpha, beta, alpha/theta, or high beta as the preferred form of treatment, while three (33.3%) chose SCP, one study (11.1%) decided on infra-low frequency EEG BFB, and one study (11.1%) did not specify the applied form of EEG BFB treatment. The most important information of all studies included in this review are summarized in Table 1 for EEG BFB, Table 2 for HRV BFB and Table 3 for EMG BFB. This consisted of the author (year), patient condition, number of subjects, control group, BFB type, type of intervention, results, and limitations. Moreover, for further clarification, Table 4 summarizes the different types of BFB used to treat various mental health conditions.

#### 3.2.1. EEG Biofeedback

##### Structural Features of EEG BFB

Nine studies examined EEG BFB, five of which included a comparative treatment, such as a placebo, another therapeutic intervention, or no treatment. Four interventions (44.4%) used a randomized study design, one (11.1%) a non-randomized, and the remaining four (44.4%) used an experimental study design. The mean number of BFB sessions per study was 21 (range 4–50), with EEG BFB exposure lasting 41.5 min (range 25–60 min) on average per session. The majority of the studies conducted between 10 and 25 sessions (66.6%). Only one study implemented fewer than 5 sessions (11.1%), whereas the remaining two studies conducted between 40 and 50 sessions (22.2%). While multiple studies did not further specify the duration of EEG BFB sessions (55.5%), a few studies conducted sessions up to 30 min (22.2%). The remaining studies implemented sessions lasting either 45 (11.1%) or 60 min each (11.1%). Most studies applied the treatment two to three times a week (44.4%).

##### Effectiveness of EEG BFB

Kopřivová et al. [25] found that alpha-regulation EEG BFB effectively reduced obsessive–compulsive disorder (OCD) symptoms, particularly anxiety. Three studies examined the therapeutic effects of different EEG BFB approaches: alpha-regulation EEG BFB for depression and anxiety [18]; alpha, beta, and sensorimotor rhythm (SMR) training for substance abuse [23]; and alpha-regulation EEG BFB for OCD [25]. All three studies reported clinical improvements following EEG BFB interventions. One study focused solely on alpha EEG BFB, showing significant reductions in subjective anxiety, as well as notable improvements in quality of life. This led to significant clinical improvement in objectively assessed anxiety, outperforming medication [18].

In contrast, three experimental studies using slow cortical potential (SCP) BFB explored its effects on psychosocial and judgmental symptoms in patients with schizophrenia [31], depression [30], and alcohol abuse [32]. These studies found SCP BFB to be ineffective in reducing clinical symptoms in any of the conditions.

Overall, regulating cortical oscillatory activity through EEG BFB seems particularly promising for conditions where inducing specific states of consciousness is key to alleviating symptoms [36].

Other studies reported improvements in depression and anxiety [18], as well as in substance abuse [23] and eating disorders [28]. However, three studies on SCP BFB showed no significant symptom reduction in schizophrenia, depression, or alcohol abuse (33.3%). Overall, EEG BFB shows potential for disorders linked to specific conscious states.

##### Participants of EEG BFB

Seven out of the nine studies included a control group in their intervention (77.7%). The majority of the studies included between 10 and 36 participants (77.7%), while the remaining studies included between 100 and 121 participants (22.2%). Several studies applied the intervention to patients suffering from depression (33.3%) or alcohol/substance use disorder (33.3%). The remaining studies focused on patient conditions such as eating disorders and post-traumatic stress disorder (PTSD) (11.1%), OCD (11.1%), and schizophrenia (11.1%).

#### 3.2.2. HRV Biofeedback

##### Structural Features of HRV BFB

Seven studies examined HRV BFB, two of which included randomized controlled trials (28.5%). One used a randomized crossover study design (14.2%). Four studies included in this review were pilot studies (57.1%). On average, patients received eight (range 3–16) sessions of HRV BFB, which each session lasting a mean of 34.6 min (range 10–75 min) per session. The majority of the studies conducted between three and eight sessions (71.4%). The remaining two studies conducted between 12 and 16 sessions (28.5%). While one study did not further specify the duration of HRV BFB sessions (14.2%), a few studies conducted sessions between 10 and 30 min (57.1%). The remaining studies implemented sessions lasting between 35 and 75 min each (28.5%). Most studies applied the treatment one to three times a week (57.1%).

##### Effectiveness of HRV BFB

Four HRV BFB studies reported a significant clinical change in symptoms [16,17,21,22]. In their study, Penzlin et al. [21] found that HRV BFB significantly reduces the perceived craving sooner than the control group in alcohol abuse patients. Furthermore, anxiety symptoms as well as cardiac autonomic and motor function could also be improved. Similarly, other studies found improvement in resting low-frequency heart rate variability and cardiorespiratory coherence in depressive patients [17]. Other studies showed improvements in stress [22] and anxiety levels [16]. The study by Scolnick et al. [33] found that patients with eating disorders, particularly anorexia nervosa, do not face an increased risk of bradycardia. These patients show higher-frequency HRV saturation compared to healthy individuals. Similarly, patients with alcohol and drug dependence [24] reported positive effects from HRV BFB, although the study did not yield results with a clear medium effect size. Eddie et al. [24] and Teeravisutkul et al. [22] both used the same HRV BFB protocol [37]. In a study by Cheng et al. [27] focusing on respiration-based BFB for schizophrenia, no significant changes were observed in anxiety levels between the clinical groups when comparing increases versus decreases in respiration. Overall, the HRV BFB studies in the review provided more detailed methodology than the other studies.

##### Participants of HRV BFB

Five out of the seven studies included a control group in their intervention (71.4%). The majority of the studies included between 35 and 70 participants (71.4%), while the remaining studies included between 15 and 25 participants (28.5%). Several studies applied the intervention to patients suffering from alcohol/substance use disorder (42.8%) or depression (28.5%). The remaining studies focused on patient conditions such as eating disorders (14.2%), and schizophrenia (14.3%).

#### 3.2.3. EMG Biofeedback

##### Structural Features of EMG BFB

Four articles outlined an EMG BFB protocol. One study was randomized (25%), two were (50%) non-randomized, and the remaining study (25%) was a pilot study. The mean number of EMG BFB sessions conducted per intervention was 9.25 (range 7–14), lasting for 41.7 min (range 30–60 min) per session on average.

The majority of the studies conducted between 7 and 14 sessions (75%). Only one study implemented between zero and eight sessions, depending on the patient (25%). While most studies conducted sessions between 30 and 40 min (50%), one study implemented EMG BFB sessions lasting 60 min (25%), and the remaining study did not further specify the duration of the sessions (25%). The majority of the studies applied the treatment one to three times a week (50%). One of the remaining studies conducted the feedback daily, while the other one did not further specify the frequency of the EMG BFB sessions.

##### Effectiveness of EMG BFB

Four articles described an EMG BFB protocol (see Table 1). One study was randomized (25%), two were non-randomized (50%), and one was a pilot study (25%). On average, participants received 8.75 EMG BFB sessions (ranging from 7 to 12), with each session lasting 36.6 min on average (ranging from 20 to 60 min). Most studies used EMG BFB to treat anxiety and related dysfunction in patients with schizophrenia. The remaining two interventions focused on depression and alcohol use. While the patients were able to modify their muscle activity, this did not reliably correlate with symptom improvement. Decreasing muscle activity was helpful in reducing general stress levels, but it was insufficient to draw conclusions about its effect on specific psychiatric symptoms. The study by Blue and Blue [34] found significant improvements in depression symptoms and suggested that EMG BFB is a valid therapeutic technique. One of the schizophrenia interventions reported an improvement in social functioning [26]. The study by Pharr and Coursey [26] also showed that EMG BFB training could be successfully used to treat chronic, hospitalized schizophrenia patients without increasing pathology or hallucinations.

##### Participants of EMG BFB

Two out of the four studies included a control group in their intervention (50%). The majority of the studies included between 30 and 40 participants (75%). The remaining study included 20 patients in their research (25%). The studies applied the intervention to patients suffering from alcohol use disorder (25%), depression (25%), psychophysiological disorders (25%), and schizophrenia (25%).

### 3.3. Subgroup Analyses

A subgroup analysis by BFB type revealed that both EEG BFB and HRV BFB interventions were similarly effective for depression and anxiety, as well as craving reduction in substance use disorders, with EEG BFB being effective in two out of three studies for substance use disorder and two out of three studies for depression and anxiety and HRV BFB being effective in two out of three studies for substance use disorder and two out of two studies for depression and anxiety.

The subgroup analysis by diagnosis revealed that most diagnoses treated with BFB were depression and anxiety (six studies) or substance use disorder (seven studies). For depression and anxiety, EEG BFB and HRV BFB were similarly effective (EEG BFB: two out of three studies; HRV BFB: two out of two studies). This also accounts for substance use disorder, for which two out of three studies revealed effectiveness for both EEG BFB and HRV BFB. However, for eating disorders and OCD, only one study each revealed effectiveness by applying EEG BFB. For schizophrenia, effectiveness has been found in three studies, with one study applying EEG BFB (1/1), one study applying HRV BFB (1/1) and one study applying EMG BFB (1/1).

### 3.4. Reported Limitations

Five of the twenty studies (25%) included in this review reported small sample size as one of the limitations in their respective studies [16,19,21,28,30]. Three studies (15%) mentioned the lack of a control group as a limiting factor of their research [16,19,33]. One study (5%) did have a control group but did not include a control intervention during the study period [17]. Another study (5%) described a lack of a controlled environment during some of the treatment, which included homework exercises [35]. Furthermore, three studies (15%) described possible limitations related to the follow-up by reporting the lack of a follow-up assessment in general [18]. Other studies have highlighted the potential for social desirability bias, as the follow-up survey was conducted after discharge. Additionally, the lack of control in follow-ups with outpatients was discussed, as it could not account for factors affecting the assessment [16,22]. The effect of medication on the results was reported by two studies (10%) [19,31]. Limited comparability as well as the extrapolation of the study results were highlighted in two studies (10%) [25,29]. Among the studies reviewed, only one study did not report any limitations to their research [34].

### 3.5. Future Directions

The researchers’ proposed future directions for research can be summarized into two main categories: recommendations for improving the methodology of future studies and suggestions for new areas of research to explore. Eight of the twenty included studies (40%) provided future directions regarding the methodology of future studies. Two of these (10%) recommended a larger, more heterogeneous sample [16,24]. Another two studies (10%) suggested alterations in the duration and an increase in the number of BFB sessions [23,29]. Furthermore, two studies (10%) recommended consistent documentation of the intervention via protocols [19,24]. One study (5%), by Cheon et al. [19], also suggested that future studies should focus on cost effectiveness as well as relapse prevention. Similarly, Teeravisutkul et al. [22] proposed conducting more follow-ups to test the persistence of BFB training. Furthermore, one study (5%) by Cheng et al. [27] recommended focusing more on differences in the functional levels of patients. Lastly, Denney et al. [20] suggested replicating their results as a basis for future research while implementing stricter controls and conditions.

Eleven of the twenty included studies (55%) provided suggestions for new areas of research to explore. Five of these studies (25%) proposed further investigation into both the short-term and long-term effects of BFB on various symptoms and disorders, such as trauma-associated avoidance, eating disorders, OCD, schizophrenia, SUD, and other psychiatric diagnoses [24,25,28,31,34]. One study (5%) by Tatschl et al. [17] recommended investigating improvements in vagal functioning through BFB interventions, as well as conducting further research on the psychological and neurobiological mechanisms underlying HRV BFB. Two studies (10%) proposed further investigation into different aspects of HRV BFB, such as disease stage-specific effects in patients with alcohol dependence or the significant difference in HRV between healthy individuals and anorexia patients as a potential biomarker for the disease [21,33]. Pharr and Coursey [26] also proposed further investigation into the mechanisms by which EMG BFB improves a patient’s social behavior. Furthermore, Winkeler et al. [28] suggested researching the supplementation of the assessment of eating disorder psychopathology with other instruments, while Scott et al. [23] proposed focusing on subjects’ observable behaviors and possible changes in future research. Lastly, Ko and Park [29] suggested measuring other variables, such as sleep disorders, anxiety, and rage. Four of the studies (20%) did not propose any suggestions for future research [18,30,32,35].

## 4. Discussion

This is the first scoping review investigating how BFB is applied and implemented in inpatient settings and whether there is evidence of its effectiveness. Based on this, if possible, recommendations for future implementation should be derived. This scoping review included 20 articles published between 1979 and 2022. Twelve of the included studies reported significant improvements in clinical symptoms for various disorders, such as OCD, depression, anxiety, substance and alcohol use disorders, schizophrenia, and eating disorders in inpatient treatment settings. Additionally, positive effects have been observed in stress reduction, as well as improvements in cardiac autonomic and motor function. Even though these results highlight the benefits of BFB in inpatient care and give an indication for the importance of an integration in treatment in inpatient mental health facilities, some of the included studies did not demonstrate significant results or even showed negative trends. However, no consensus on the structural features of BFB training has been established across the studies included in this review. The duration of BFB sessions (EEG BFB, HRV BFB, and EMG BFB) varied widely, ranging from 10 to 75 min, with sessions being administered one to seven times per week. This variability was also reflected in the total number of BFB sessions completed by patients, which ranged from one to 50 sessions over the course of their treatment. The absence of a standardized BFB protocol is further evident in the differing methodologies across studies: some relied on sessions conducted in a controlled environment (i.e., at the treatment facility) and led by trained staff, while others incorporated homework exercises that the patients completed independently. This lack of consistency in the application of BFB has been identified by several of the reviewed studies as a limitation of their research. Eight of the included studies did not demonstrate significant improvements in clinical outcomes. This could have various reasons, such as methodological weaknesses like small sample sizes, a lack of control groups, or insufficient randomization and blinding. Moreover, it is not always clear which type of BFB is effective for treating a specific disorder. Although BFB has been researched for several decades, there are still no standardized guidelines on how the training should be conducted. The present scoping review aims to provide an overview of the existing literature and derive recommendations for the implementation of BFB.

The results of this scoping review show that different studies conduct BFB in a highly heterogeneous manner. The implementation of BFB varied significantly in terms of the number of sessions, session duration, as well as the disorders treated and parameters trained. Similar observations have been made in outpatient settings. There are significant differences in the implementation of BFB not only in inpatient settings but also in outpatient contexts. These differences pertain to the number of sessions, session duration, and frequency, as well as the training of individual parameters. For instance, Walker et al. [38] conducted neurofeedback (NFB) for depressive disorders with 6 sessions of 20 min each, while Yu et al. [39] performed an average of 20 sessions, each lasting 30 min, for the same condition. Similarly, in the treatment of eating disorder-related symptoms in obese patients, the number and duration of sessions vary, from four sessions of 9 min each [40], to eight sessions of 21 min each [41], to ten sessions lasting 30–45 min each [41]. Moreover, recent trials validate EEG BFB efficacy in chronic pain management and conducted 12 sessions [10], whereas another study applied 5 sessions and achieved significant reduction of post-COVID-related symptoms such as depression and anxiety [42]. Determining the actual treatment effect of NFB is challenging, and effect sizes are rarely reported, making the conduct of a meta-analysis difficult.

The studies included in this scoping review are difficult to compare due to the heterogeneity in their implementation, which limits the ability to draw conclusions regarding clinical implications. However, most of the included articles reported significant improvements after BFB interventions, suggesting that the application of BFB may lead to a desired outcome independently of a specific training regimen. Nevertheless, due to the limited comparability, it is challenging to derive recommendations for the application of BFB. Therefore, it might be important to establish standardized criteria for implementation. An initial attempt to achieve this within NFB training has already been made through the CRED-nf checklist [43]. This checklist provides guidelines for reporting NFB studies, ensuring that all relevant data necessary for replication of the study are reported. However, not only do methodological decisions play a significant role in the effectiveness of BFB but also the perceived self-efficacy [44]. This is influenced not only by the patient’s subjective sense of control or success in training but also by the interaction with the therapist. Unfortunately, there are no current guidelines or recommendations for addressing this effect to make studies more comparable. However, the heterogeneity across NFB studies makes the usage of standardized reporting practices, such as CRED-nf [43] or Consort [45], in future studies even more necessary. An initial attempt to address this issue was made by this research group. Schmidt et al. [46] developed a manualized NFB training and tested it in an outpatient setting. Preliminary results indicate that the acceptance of the manualized training was satisfactory for both patients and NFB practitioners [47]. Furthermore, the patients were satisfied with the training. Results regarding its influence on psychometric data, as well as its applicability and efficacy in inpatient settings, are still pending.

The included studies have identified several limitations that should be considered when interpreting their results. Primarily, limitations concerning methodology, such as small sample sizes, the lack of control groups, etc., were mentioned. Unfortunately, none of the included studies described difficulties in the implementation of BFB, based on which recommendations for future implementation could be derived. This would be particularly important in relation to the implementation of BFB in inpatient settings, as it significantly differs from outpatient settings and faces challenges that could influence or even impair the training. Schmidt et al. [11] showed that implementation difficulties are primarily caused by hectic routines, unforeseen events on the ward, staff shortages, and a tightly scheduled therapy calendar. However, further studies are needed to examine the implementation of BFB in inpatient contexts and to also report emerging challenges.

At this point in time, it is difficult to derive recommendations for BFB training that are effective in treating mental disorders in inpatient settings. Currently, a variety of different training methods are being conducted, which lead to varying degrees of symptom relief. The facilities conducting BFB in their inpatient routine seem to achieve positive outcomes, despite using different approaches. This raises the question of the need for standardized criteria. However, for the benefit of the patients, such standardization may not be necessary. Nevertheless, the studies provide indications that would be advisable for the future implementation of BFB. For instance, they point to a sufficiently high number of sessions, although it is currently challenging to establish a specific number, as studies with a lower number of sessions also yielded significant results. For example, studies implementing ≥10 sessions (e.g., Scott et al. [23]; Ko & Park [29]) demonstrated sustained effects, suggesting this as a minimal threshold for clinical adoption. Moreover, the included studies show effects from BFB intervention with an average duration of 39.6 min, which can be considered as a guideline for session length. Furthermore, a consistent documentation of sessions should be conducted, and follow-up sessions should be implemented to examine the sustainability of the effects achieved through BFB.

### Limitations

This work has certain limitations, which are common in scoping reviews. First, publication bias need to be considered, as studies with non-significant findings may not have been published, and many studies have not been reproduced to verify their results. To address this issue, future meta-analyses should conduct specific analyses, for example funnel plots or z-curves. The number of studies included may have been restricted due to incomplete data reporting. Second, selection bias could potentially affect the inclusion of relevant studies. However, we aimed to minimize this by having two of the authors independently assess the studies. Third, it is important to note that only the reference lists of the included studies were examined, meaning that some studies cited in the excluded articles may have been overlooked. Fourth, no quality appraisal was applied to the included studies, which limits the interpretation of the results. What should be taken into account is that only two out of twelve studies reporting significant effects provide effect sizes. This suggests a lack of study quality, which means the results should be interpreted with caution. Future reviews or meta-analyses should apply a proper quality appraisal.

## 5. Conclusions

This scoping review identified 20 studies on the application of BFB in inpatient treatment settings, aiming to provide an overview of this field. The implementation of BFB varied significantly across the studies, with a broad range in terms of the number of sessions, session duration, session frequency, and the disorders treated with predominantly small sample sizes. Most of the included studies were able to demonstrate benefits of the BFB intervention in treating clinical symptoms. Due to the heterogeneity of the studies, they can only be compared to a limited extent, making it difficult to derive general recommendations for the ideal and future implementation of BFB. Nevertheless, the question remains open to what extent a methodologically consistent approach is necessary to achieve clinical success.

## Figures and Tables

**Figure 1 jcm-14-03491-f001:**
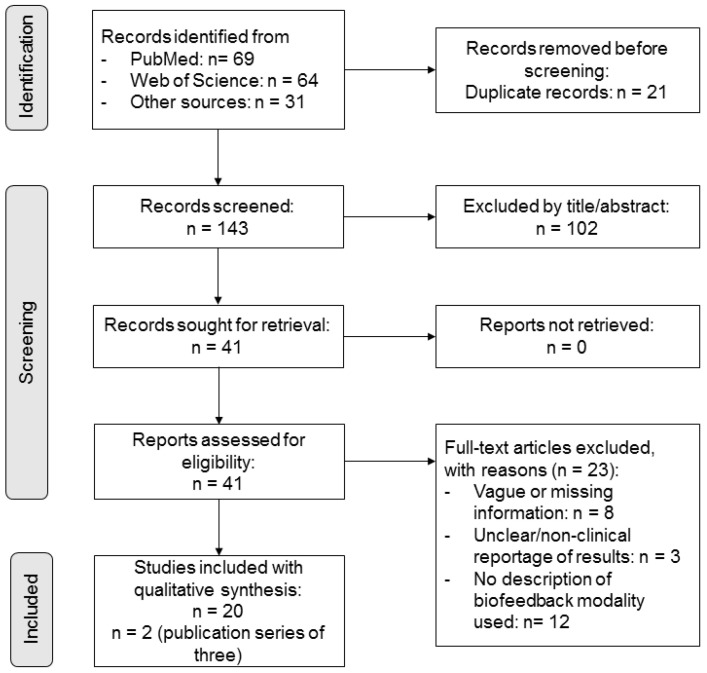
Flow chart of the study search and selection process.

**Figure 2 jcm-14-03491-f002:**
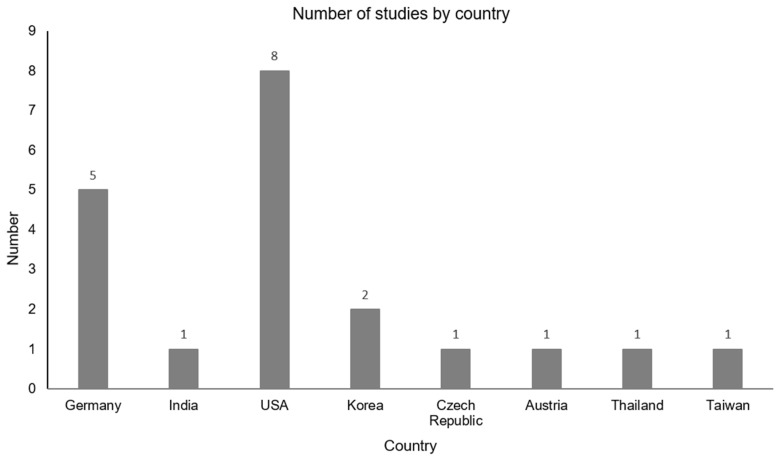
Distribution of included studies by country. Most studies were published in the USA and Germany.

**Figure 3 jcm-14-03491-f003:**
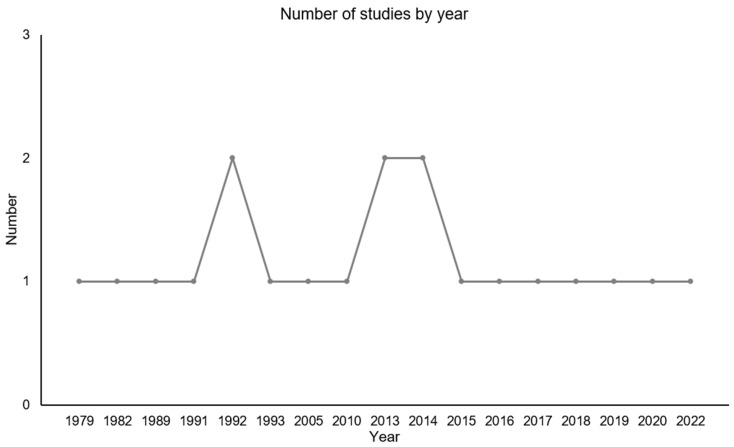
Distribution of included studies by year. Most studies were published in 1992, 2013, and 2014.

**Table 1 jcm-14-03491-t001:** Summary of EEG BFB studies included in this scoping review.

Author (Year)	Patient Condition	Number of Subjects	Control Group	BFB Type	Type of Intervention	Manualized Training	Results	Limitations
Winkeler et al. (2022) [28]	Eating disorder and post-traumatic stress disorder	*n* = 18	*n* = 18	Infra-low frequency EEG	12 sessions 30–40 min 2×/week	No	EG significantly improved in restrained eating, more weight gain, reduced avoidance behavior, and less complications in course of treatment.	Lack of homogeneity in diagnoses; small sample size
Bhat (2010) [18]	Anxiety and depression	*n* = 50	*n* = 50	Alpha EEG	40 sessions 5×/week	No	Improvement of anxiety symptoms: medication showed higher improvement than BF; in EG, women showed higher improvement with BF than men;	Lack of a follow-up to assess continued benefit
Scott et al. (2005) [23]	Substance use disorder	*n* = 60	*n* = 61	Beta EEG and SMR	40–50 sessions 45 min 4–5×/week 2×/day	No	significantly more dropout in CG, EG significantly longer abstinent, and significant improvement in TOVA Questionnaire and in 5/10 MMPI Scales.	Relied on self-report for abstinence check
Cheon et al. (2016) [19]	Depression	*n* = 20	*n* = 0	Beta and alpha/theta EEG	16–24 sessions 60 min 2–3×/week	No	Significant improvement in depressive symptoms, anxiety, and clinical illness; increased remission and response rates.	Small sample size, lack of control group, non-blinding subjects; patients received medication
Ko & Park (2018) [29]	Alcohol use disorder	*n* = 17	*n* = 19	Alpha EEG and high-beta training	10 sessions 40 min 2–3×/week	No	Significant increase in basic psychological need satisfaction, alcohol abstinence self-efficacy, and self-regulation in EG, with no significant increase in alpha waves and decrease in high-beta waves; significant increase in high beta of CG.	Lack of comparability; difficult to extrapolate the results to patients with alcohol use disorder
Kopřivová et al. (2013) [25]	Obsessive–compulsive disorder	*n* = 10	*n* = 10	EEG and sham feedback	18 sessions 25–30 min 3×/week	No	NFB group showed significantly higher percentage-based reduction in compulsions.	Limited spatial specificity; results limited to specific phenotype
Schneider et al. (1992a) [30]	Depression	*n* = 8	*n* = 8	SCP	20 sessions 5×/week	No	SCP self-regulation impairment specific for schizophrenic patients, with no comparable deficits found for patients with depression.	No significant self-regulation of SCP in the control group, contradicting other study findings; small sample size
Schneider et al. (1992b) [31]	Schizophrenia	*n* = 12	*n* = 12	SCP	20 sessions on 20 consecutive days	No	After direct feedback, schizophrenic patients were able to regulate SCP systematically compared to patients with alcohol dependence. The schizophrenia EG was unable to achieve a transfer performance of self-regulation for SCP; a correlation was found between the inability to regulate SCP and the duration of the illness.	Observed group effect may be more related to the use of medication than to schizophrenia
Schneider et al. (1993) [32]	Alcohol use disorder	*n* = 10	*n* = 0	SCP	4 sessions on 4 consecutive days	No	The greatest increase in SCP differentiation is expected in the transfer condition with increasing abstinence; learning self-regulation of SCP with feedback takes more time compared to the four sessions on four consecutive days.	The information processing of the feedback stimulus may have prevented the modification of the SCP

Notes: EG = experimental group, CG = control group, and SCP = slow cortical potential.

**Table 2 jcm-14-03491-t002:** Summary of HRV BFB studies included in this scoping review.

Author (Year)	Patient Condition	Number of Subjects	Control Group	BFB Type	Type of Intervention	Manualized Training	Results	Limitations
Beckham et al. (2013) [16]	Depression and anxiety	*n* = 15	*n* = 0	HRV	8 sessions 30–60 min 2×/week	No	STAI, WEMWBS, and LASA showed significant improvements at time points A and B compared to baseline.	Lack of control group; results only on short-term effects; small sample size; follow-up survey after discharge: social desirability bias
Eddie et al. (2014) [24]	Substance use disorder	*n* = 21	*n* = 20	HRV	3 sessions 60–75 min 1×/week	No	Treatment + BFB: greater effect on craving reduction than CG but not significant; negative correlation of HRV and stress interaction: HRV at beginning of treatment only predicts change in craving in CG; high HRV is connected to higher reduction in craving between begin and end of treatment.	Lack of a significant overall effect of HRV BFB despite a mean effect size reduction in abstinence due to the short duration of the training
Penzlin et al. (2015) [21]	Alcohol use disorder	*n* = 24	*n* = 24	HRV	6 sessions 20 min 3×/weeks	No	BFB group: perceived reduction in craving sooner than control; decrease in anxiety (vs. control); improved cardiac autonomic function; improved vasomotor function after completion.	Late-stage ethyl-toxic damage to cardiac autonomic fibers may have reduced HRV BFB responsivity; small sample size; Laser Doppler flowmetry insensitive to individual vasomotor dysfunction
Scolnick et al. (2014) [33]	Eating disorder	*n* = 24	*n* = 0	HRV	12 sessions 10 min 5–7×/week	No	HRV BFB training is safe in this population; can be used alongside yoga and meditation.	Lack of control group
Tatschl et al. (2020) [17]	Depression	*n* = 34	*n* = 34	HRV	5 sessions 35 min 1×/week	No	Larger recovery in depressive symptoms than CG (but decreased in follow-up), as well as increases in resting low-frequency HRV and cardiorespiratory coherence.	No assessment of symptoms between post-assessment and follow-up (12 months); no assessment of slow breathing training between post-intervention and follow-up; potential placebo effect; CG did not get additional control intervention
Teeravisutkul et al. (2019) [22]	Alcohol use disorder	*n* = 17	*n* = 18	HRV	16 sessions 30 min 4×/week	No	EG: decreased stress and craving after training and 1-month follow-up, CG only immediately after training; higher difference in craving and stress scores at baseline and post-intervention than CG.	Follow-up performed on outpatients, so no proper control for factors that could affect follow-up results
Cheng et al. (2017) [27]	Schizophrenia	*n* = 30	*n* = 30	HRV, EMG, GSR, and RR	6 sessions 2×/week	No	Significant improvement in anxiety (EG); significant decrease in HR and RR (EG); HADS score and anxiety subscores decreased significantly as number of interventions increased.	Highly functioning chronic schizophrenic patients only

Notes: EG = experimental group; CG = control group.

**Table 3 jcm-14-03491-t003:** Summary of EMG BFB studies included in this scoping review.

Author (Year)	Patient Condition	Number of Subjects	Control Group	BFB Type	Type of Intervention	Manualized Training	Results	Limitations
Blue & Blue (1979) [34]	Depression	*n* = 30	*n* = 10	EMG	14 sessions 30–40 min on 14 consecutive days	No	(In individual sessions 6, 7, and12) muscle tension lower in manic and agitated group (vs. depressed and comparative group).	No limitations mentioned
Pharr & Coursey (1989) [26]	Schizophrenia	*n* = 10	*n* = 20	EMG	7 sessions 30 min 3×/week	No	Significant lower muscle tension in EG; no increase in psychopathy in EG patient.	Possible practice effect in Finger-Tapping Test; no balancing of negative symptoms of chronic schizophrenia
Denney et al. (1991) [20]	Alcohol use disorder	*n* = 20	*n* = 0	EMG and thermal BFB	0–8 sessions	No	0–5 group: no significant difference to no training; 6–7 group and 8+: significantly better than no training and 0–5 group; strongest effect in 8+ group at 3-month mark; abstinence decreases after 6 months post-discharge but slower in the BFB group.	Retrospective pilot study: not all variables that should be considered in a formal research format were examined
Ford et al. (1982) [35]	Psychophysio-logical disorders	*n* = 37	*n* = 0	EMG and thermal BFB	8 sessions 60 min 1×/week	No	Lower MMPI scores = the lower the initial burden of mental illness of the inpatients; younger patients (17–24 years of age) were unsuccessful in therapy (due to lack of adherence); patients older than 30 years achieved no effect in only 21% of subjects with lower levels of distress.	Patients learn and implement BFB at different speeds; the familiar, quiet environment and the support provided by the clinic staff in the inpatient setting was not guaranteed during the homework exercise

Notes: EG = experimental group; CG = control group.

**Table 4 jcm-14-03491-t004:** Overview of types of BFB used to treat different mental health disorders.

	EEG BFB (N)	HRV BFB (N)	EMG BFB (N)
Depression and anxiety	3 (33.3%)	2 (28.6%)	1 (25%)
Substance use disorder	3 (33.3%)	3 (42.9%)	1 (25%)
Eating disorder	1 (11.1%)	1 (14.3%)	0 (0%)
Schizophrenia	1 (11.1%)	1 (14.3%)	1 (25%)
Obsessive–compulsive disorder	1 (11.1%)	0 (0%)	0 (0%)
Other	0 (0%)	0 (0%)	1 (25%)

Notes: EEG BFB = electroencephalographic biofeedback, HRV BFB = heart rate variability biofeedback, and EMG BFB = electromyography biofeedback.

## Data Availability

The original contributions presented in this study are included in the Appendix A. Further inquiries can be directed to the corresponding author.

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
