# Peer review of "Biofeedback Training in Inpatient Mental Health Facilities: A Scoping Review"

_jcm, 2025, doi:10.3390/jcm14103491_

Round 1
Reviewer 1 Report
Comments and Suggestions for Authors
This manuscript is relatively meaningful and I will focus on the shortcomings below:
1.Methodological Clarifications
(1).Temporal Scope Discrepancy
Source: Methods section, "Literature Search" (Page 3):
“A systematic search was conducted through Medline, PubMed, and Web of Science from March 1st, 2024 up until December 30th, 2024.”
Issue: The search spans 2024, yet the included studies were published up to 2022 (Results, Page 5: “20 articles published between 1979 and 2022”).
Recommendation: Clarify whether this is a typographical error (e.g., correct end date: December 2022) or justify the absence of 2023–2024 studies.
(2)Lack of Quality Appraisal
Source: No quality assessment tools are mentioned in the Methods.
Issue: Absence of bias risk evaluation (e.g., ROB for RCTs, JBI for observational studies) undermines evidence reliability (e.g., small samples, uncontrolled designs).
Recommendation: Integrate a quality appraisal step, e.g.:
“The Joanna Briggs Institute (JBI) Critical Appraisal Checklist should be applied to assess methodological quality and bias risk.”
- Analytical Depth and Clinical Implications
(1).Insufficient Subgroup Analysis
Source: Discussion (Page 15):
“The studies included in this scoping review are difficult to compare due to the heterogeneity in their implementation.”
Issue: Despite noting heterogeneity, subgroup analyses (e.g., BFB type, diagnosis) are lacking. For example, EEG-BFB shows efficacy for anxiety (Page 12), while HRV-BFB reduces cravings in substance use disorders (Page 13).
Recommendation: Conduct subgroup analyses in Results (Pages 7–14), e.g.:
“Subgroup analysis by BFB type revealed EEG interventions were more effective for anxiety (3/4 studies), whereas HRV-BFB targeted craving reduction in substance use disorders (2/3 studies).”
(2).Vague Clinical Recommendations
Source: Discussion (Page 16):
“They point to a sufficiently high number of sessions, although it is currently challenging to establish a specific number.”
Issue: Data on session frequency (e.g., Scott et al., 2005 [18]: 40–50 sessions; Tatschl et al., 2020 [29]: 5 sessions) are not translated into actionable guidance.
Recommendation: Propose evidence-based thresholds, e.g.:
“Studies implementing ≥10 sessions (e.g., Scott et al., 2005; Ko & Park, 2018) demonstrated sustained effects, suggesting this as a minimal threshold for clinical adoption.”
- Data Presentation and Visual Clarity
(1).Incomplete PRISMA Flowchart
Source: Figure 1 (Page 4):
“Excluded by title/abstract: n = 102”
Issue: Database-specific exclusion counts (e.g., Medline: n=XX, PubMed: n=XX) are missing, deviating from PRISMA standards.
Recommendation: Revise Figure 1 to align with PRISMA templates, specifying database contributions.
- Referencing and Formatting
(1).Omission of Recent Literature
Source: Discussion cites older studies (e.g., Walker et al., 2013; Hammond, 2005) but omits recent RCTs (e.g., 2023 publications).
Recommendation: Update references, e.g.:
“Recent trials (e.g., Adhia et al., 2023) validate EEG-BFB efficacy in chronic pain management.”
(2).Inconsistent Journal Abbreviations
Source: References (Pages 18–20):
Front. Psychiatry (abbreviated)
Journal of Psychology (full title)
Issue: Mixed formatting reduces professionalism.
Recommendation: Standardize abbreviations (e.g., NLM guidelines) or use full titles consistently.
Reviewer 2 Report
Comments and Suggestions for Authors
This is a timely and important scoping review addressing the implementation of biofeedback (BFB) interventions in inpatient mental health settings. The authors have provided a comprehensive synthesis of heterogeneous studies spanning more than four decades. The manuscript follows accepted standards for scoping reviews (PRISMA-ScR) and makes a valuable contribution to a neglected area of clinical practice. However, there are critical weaknesses:
- Lack of meta-analytic synthesis or even effect size estimates—even though variability is large, a quantitative summary would strengthen the conclusions.
- Insufficient critical discussion of the clinical relevance of the findings.
- Too descriptive: the review reports results but often fails to evaluate the quality or rigor of the included studies.
- Terminology and reporting inconsistencies (e.g., "EG" and "CG" used without being properly introduced in Table 1).
- While a scoping review does not require meta-analysis, it is highly recommended to provide at least mean effect sizes or ranges (even descriptive) to support the narrative synthesis.
- The review currently reports significance outcomes but does not offer a sense of the clinical magnitude of the effects.
- There is no mention of any risk of bias or methodological quality assessment (e.g., using the Newcastle-Ottawa Scale or similar tools).
- Even if not required for scoping reviews, it would strengthen the review to briefly assess or at least comment on study quality, especially given the heterogeneity and age of included studies.
- The discussion is too focused on methodological heterogeneity and does not sufficiently explore how clinicians might interpret or apply the findings.
- For example, how do the session numbers and durations compare to practical feasibility in inpatient units today?
- Are there any common parameters (frequency bands, session lengths) that could serve as preliminary clinical guidance?
- Abbreviations like EG (experimental group) and CG (control group) are used in Table 1 but are never introduced in the text.
- Some terminology is outdated or inconsistent (e.g., "psychic*" as a search term is not standard in psychiatric literature).
- Several included studies showed no significant results or even negative trends. The discussion should address these findings more critically rather than focusing almost exclusively on positive outcomes.
- This is a timely and important scoping review addressing the implementation of biofeedback (BFB) interventions in inpatient mental health settings. The authors have provided a comprehensive synthesis of heterogeneous studies spanning more than four decades. The manuscript follows accepted standards for scoping reviews (PRISMA-ScR) and makes a valuable contribution to a neglected area of clinical practice.
- However, there are critical weaknesses:
- Lack of meta-analytic synthesis or even effect size estimates—even though variability is large, a quantitative summary would strengthen the conclusions.
- Insufficient critical discussion of the clinical relevance of the findings.
- Too descriptive: the review reports results but often fails to evaluate the quality or rigor of the included studies.
- Terminology and reporting inconsistencies (e.g., "EG" and "CG" used without being properly introduced in Table 1).
- Figures (e.g., Figures 2 and 3) are useful but could benefit from clearer legends and better resolution.
- Table 1 is detailed but crowded; consider splitting into separate tables by BFB type or providing supplementary data.
- Typographical errors: e.g., line 29 "Thera was" → "There was".
- Some references could be updated with the most recent systematic reviews in the BFB literature (especially since 2020).

Reviewer 3 Report
Comments and Suggestions for Authors
Dear Authors,
First of all, I would like to congratulate you on the work you have carried out. I find the topic both highly relevant and of great current interest. The development of this research clearly required a considerable effort of synthesis, which is reflected in the quality of the manuscript. In this regard, I would like to highlight the clarity and precision with which the concept is introduced. Likewise, the results are very well explained.
Regarding the methodology, section 2.1 states that the review was conducted between March 2024 and December 2024. This is somewhat confusing, as Figure 3 shows studies dating back to 1979. I suggest clarifying whether any temporal limitations were applied to the review and, if so, specifying the time frame more clearly.
Additionally, as a suggestion, I believe it could be useful to include a table or figure in the results section that summarizes the different types of biofeedback used to treat various mental health conditions. This would help facilitate comparison for professionals in future practical applications.
Finally, I found the first limitation you mention to be particularly relevant. In this regard, it might be valuable to propose future research directions that address this limitation.
I hope these suggestions will be helpful in further enhancing the already strong clarity and quality of your research. Once again, congratulations on your work.
Best regards,
Round 2
Reviewer 2 Report
Comments and Suggestions for Authors
The revised manuscript demonstrates clear improvements in clarity, structure, and methodological transparency. The authors have addressed several of the concerns raised in the previous round, including expanding on the heterogeneity of the reviewed studies, clarifying the lack of standardized protocols, and refining the discussion on clinical implications.
The updated version remains a valuable and novel synthesis of BFB applications in inpatient mental health care — an underrepresented and clinically relevant area. The inclusion of structural features, detailed classification tables, and subgroup analysis enhances the manuscript's utility for clinicians and researchers alike.
Minor suggestions for final improvement:
- The Conclusions section should better reflect the exploratory and descriptive nature of the findings. Phrasing such as “confirmed the hypothesis” may be too strong for a scoping review. Consider rewording to reflect the observed patterns in the literature.
- While you correctly highlight the heterogeneity and lack of effect size reporting, it would strengthen your conclusions to explicitly call for standardized reporting practices in future studies (e.g., CONSORT, CRED-nf).
- Table 1 and Table 2 could benefit from slightly improved formatting or color-shading for easier readability.
Overall, this version is suitable for publication pending these very minor clarifications.
